# Wound-Healing Effect of *Antheraea pernyi* Epidermal Growth Factor

**DOI:** 10.3390/insects13110975

**Published:** 2022-10-24

**Authors:** Yu-Lan Piao, Chun-Yang Zhang, Yue Zhang, Kun Qian, Ying Zhou, Jun-Yan Liu, Young-Cheol Chang, Hoon Cho, Dubok Choi

**Affiliations:** 1School of Food Engineering, Jilin Agriculture Science and Technology University, Jilin 132109, China; 2Jilin Province Sericultural Scientific Research Institute, Jilin 132012, China; 3Jilin Province Aikangshou Biotechnology Co., Ltd., Jilin 132012, China; 4Course of Chemical and Biological Engineering, Division of Sustainable and Environmental Engineering, Muroran Institute of Technology, Muroran 050-8585, Japan; 5Department of Biochemical & Polymer Engineering, Chosun University, Gwangju 61452, Korea; 6Faculty of Advanced Industry Convergence, Chosun University, Gwangju 61452, Korea

**Keywords:** wound healing, *Antheraea pernyi*, epidermal growth factor, prokaryotic expression, re-epithelialization

## Abstract

**Simple Summary:**

We performed the sequence analysis, cloning, and prokaryotic expression of cDNA from the ApEGF gene, examined the transcriptional changes, and investigated the wound-healing effect of this protein in cells and rat epidermis to evaluate the wound-healing effect of Antheraea pernyi epidermal growth factor (ApEGF). The gene sequence fragmesnt of ApEGF was 666 bp in length, encoding 221 amino acids, with a predicted protein mass of 24.19 kD, an isoelectric point of 5.15, and no signal peptide sequence. ApEGF was truncated and then subjected to prokaryotic expression, isolation, and purification. Truncated ApEGF was used for wound-healing experiments in vitro and in vivo. The results showed that after 48 h, transforming growth factor (TGF)-β1 had 187.32% cell growth, and the ApEGF group had 211.15% cell growth compared to the control group in vitro. In rat epidermis, truncated ApEGF showed a significantly better healing effect than the control.

**Abstract:**

To evaluate the wound-healing effect of *Antheraea pernyi* epidermal growth factor (ApEGF), we performed the sequence analysis, cloning, and prokaryotic expression of cDNA from the ApEGF gene, examined the transcriptional changes, and investigated the wound-healing effect of this protein in cells and rat epidermis. Primers were designed based on available sequence information related to the ApEGF gene in a public database, and part of the ApEGF sequence was obtained. The full-length cDNA sequence of ApEGF was obtained using inverse PCR. The gene sequence fragment of ApEGF was 666 bp in length, encoding 221 amino acids, with a predicted protein mass of 24.19 kD, an isoelectric point of 5.15, and no signal peptide sequence. Sequence homology analysis revealed 86.1% sequence homology with *Bombyx mori*, 92.7% with *Manducal sexta*, 92.6% with *Trichoplusia ni*, and 91.8% with *Helicoverpa armigera*. ApEGF was truncated and then subjected to prokaryotic expression, isolation, and purification. Truncated ApEGF was used for wound-healing experiments in vitro and in vivo. The results showed that after 48 h, transforming growth factor (TGF)-β1 had 187.32% cell growth effects, and the ApEGF group had 211.15% cell growth compared to the control group in vitro. In rat epidermis, truncated ApEGF showed a significantly better healing effect than the control. This result indicated that ApEGF, which exerted a direct wound-healing effect, could be used in wound-healing therapy.

## 1. Introduction

The skin is an organ that protects the human body, and damage to this organ can lead to dangerous situations. Upon the formation of skin defects, such as trauma, skin damage, and burns, efforts are made to promote the regeneration of skin tissue and minimize traces of regenerated tissue [1]. Therefore, rapid treatment is required to recover damaged skin tissues, and wound healing using appropriate wound treatments is essential. With the development of several medicines, research on wound healing has been actively conducted [2]. However, although many chemical composites have been developed for tissue repair and wound healing, various studies are underway to develop natural and biological materials with low side effects and excellent efficacies, as long-term action is impossible owing to resistance.

For example, Kim et al. manufactured a gel that facilitates transdermal delivery using curcumin isolated from *Curcuma longa* and examined its usability for wound healing. Their results showed that the gel shortens the wound closure period and accelerates cell regeneration, suggesting its effectiveness for wound healing [3]. The wound-healing effects of *Astragali Radix* extracts on experimental open wounds in rats have also been reviewed. In the initial process, the inflammatory response is suppressed, leading to rapid re-epithelialization, and collagen fibers proliferate to promote the formation of cell substrates, helping to relocate connective tissue and contract wounds [4]. The hexane extract of *Dendropanax morbifera* Leveille was found to inhibit 15-hydroxyprostaglandin dehydrogenase to increase prostaglandin E2 synthase levels and is thus effective for wound healing [5]. Lim [6] reviewed the effects of red ginseng ginsenoside and red ginseng extract on wound healing in a diabetic wound model and found that they could be potential therapeutic agents for alleviating diabetic wounds by generating and stimulating factors involved in wound healing. Yoon [7] investigated cell proliferation, migration, collagen synthesis, intracellular signaling, and sprout outgrowth in keratinocytes to identify the effect of *Hibiscus syriacus* L. flower on skin regeneration and wound healing, and the results suggested its potential as a skin treatment agent and cosmetic raw material. The action mechanism of the hot water extract of *Acai berry* on wounds induced in the skin and oral mucosa was studied. The extract led to an increase in cell mobility, which was related to an increase in fibronectin and a decrease in the matrix metalloproteinase-1(MMP-1) gene. In addition, the expression of type-I collagen, vascular endothelial growth factor, fibronectin, and tumor necrosis factor alpha genes was increased, whereas the expression of MMP-1 and interleukin-1β genes was decreased [8]. Oh et al. [9] reviewed the wound-healing effect of callus extracts obtained from *Artemisia annua* Linne culture broth. In particular, they confirmed that the expression of cyclooxygenase-2 was reduced by more than 50%, and that wound healing was increased by approximately 70%. Thus, *A.*
*annua* callus extract has been proposed as a natural and eco-friendly agent with anti-inflammatory and wound-healing effects. Han et al. [10] reported that in hamsters with diabetes, in which natural healing is delayed, wound healing and cell proliferation were improved by recombinant human granulocyte-macrophage colony-stimulating factor from rice cells. Yun et al. [11] investigated the wound-healing effect of polydeoxyribonucleotide (PDRN) isolated from salmon milt. The application of PDRN accelerated wound contraction and significantly reduced the wound area. These results indicated the potential of PDRN as a functional material for skin regeneration in the cosmetics and medical industries. Choi et al. investigated alginic acid and chitosan hydrogel patches containing a beta-glucan nanoemulsion on the antibacterial activity for wound-repair effects. The results indicated that it can contribute to the development of a wound-healing agent with antibacterial effects as natural polymer material [12].

Epidermal growth factor (EGF) is a growth factor derived from single cells, macrophages, and platelets [13], and it plays an important role in various physiological processes [14]. It promotes the proliferation of skin, stomach, lung, tracheal epithelial, and corneal epithelial tissues, accelerates the repair of injured epidermal cells and gastric ulcers, and inhibits gastric acid secretion [15]. EGF can significantly promote the proliferation of fibroblasts as well as the migration and differentiation of epithelial cells in the wound-healing process, thereby promoting the re-epithelialization of granulation tissue and wounds [16]. Many researchers have investigated whether EGF can shorten wound-healing time. Epstein et al. suggested that EGF increased tensile strength and decreased adverse tissue events, such as failure to produce extracellular matrix proteins in saliva and the severity of oral mucositis during oropharyngeal radiation therapy [17]. Combined treatment with erythropoietin and recombinant human EGF on full-thickness wound healing in a diabetic rat model significantly improved healing time by 50% compared with EGF alone. This result indicated that the combination treatment improved wound healing, possibly through the synergistic action of both growth factors [18]. To evaluate bifunctional fusion proteins as epidermal cell-healing agents, abundant proteins in epidermal cells and fused human EGF to the collagen-binding domain of *Escherichia coli* were examined for collagen-binding activities, cell proliferation, and phosphorylation signals [19]. Kang et al. [20] developed an EGF via the fusion of the skin macromolecule transduction domain (MTD) to improve the percutaneous transmission efficiency of EGF, and the resultant EGF showed excellent permeability to the dermis. These results show that MTD-EGF efficiently improves the existing physical percutaneous transmission method through percutaneous transmission as a raw material for cosmetics containing various active substances.

*Antheraea pernyi*, a commercially cultivated silk moth and a source of food containing high-quality protein consumed in China, India, and Korea, is known for building high-quality cocoons that can be used as fibers [21]. It is also attracting attention as a new biomaterial because it has been found to have properties suitable for biological applications, such as cell culture and tissue engineering support [22]. Han et al. [23] examined the effect on skin fiber sub-cells and inflammatory cell proliferation by separating silk protein granules in various molecular weight ranges from the silk protein of *A. pernyi.* Um et al. [24] investigated the effect of the dietary feeding of silk fibroin powder from *A. pernyi* on antioxidant defense status and lipid metabolism. This result suggests that it may be beneficial as a functional biomaterial for the development of therapeutic agents against high-fat-diet-induced hyperlipidemia and related diseases. Shuai et al. designed the biomineralization process induced by prenuclear calcium and phosphorus nanoclusters to nucleate and generate hydroxyapatite crystals on the surface of the A. pernyi fibroin membrane. This result significantly promoted the osteogenic differentiation of MSCs in the absence of an osteogenic inducer and demonstrated that bone-related matrix proteins are highly expressed [25]. Polydopamine-coated *A. pernyi* silk fibroin films promoted cell adhesion and wound healing in skin tissue repair [26]. Xin et al. [27] analyzed differentially expressed genes and identified them in different functional databases. These results enrich the information in *A. pernyi* gene databases and provide insights into the potential antioxidant mechanisms of *A. pernyi*. Although there have been several studies on the biological functional properties of *A. pernyi*, there are no reports regarding the wound-healing effect of *A. pernyi* epidermal growth factor (ApEGF).

In this study, to evaluate the effect of ApEGF on wound-healing activity, the sequence acquisition, cloning, transcriptional changes, and prokaryotic expression of cDNA from the EGF gene and wound healing in cells and rat epidermis were carried out.

## 2. Materials and Methods

### 2.1. cDNA Synthesis and PCR Amplification

Total RNAs were extracted from the fifth instar larvae of *A. pernyi* using a total RNA isolation kit (QIAGEN, Valencia, CA, USA), according to the manufacturer’s instructions, and reverse transcription was performed by using a cDNA synthesis kit (Takara Bio, Shiga, Japan). PCR was performed using synthetic cDNA as a template and amplification primers. Primer sequences were designed based on the *Bombyx mori* pro-epidermal growth factor gene, as shown in Table 1. The primers used for amplification were forward 5′-GCGGTCATGTCATCGCCTTTCG-3′ and reverse 5′-GTCCTTGAACTC GCACCGC-3′. The PCR amplification conditions consisted of an initial denaturation at 94 °C for 1 min, followed by 30 cycles of denaturation at 94 °C for 45 s, 55 °C for 30 s, and extension at 72 °C for 1 min, and a final extension step at 72 °C for 5 min. PCR products were separated by 1% agarose gel electrophoresis, stained with ethidium bromide, and photographed using a video documentation system (Chemi XRS Gel Documentation System; Bio Rad, CA, USA). The PCR products were cloned into the pGEM-T easy vector (Promega Corporation, Madison, WI, USA).

### 2.2. Inverse PCR

Genomic DNA from the fifth instar larvae of *A. pernyi* was purified using a Wizard Genomic DNA purification Kit (Promega Corporation, Wisconsin, USA). The purified genomic DNA was digested with PCiI (New England Biolabs, MA, USA) at 37 °C and circularized by overnight ligation at 16 °C using T4 DNA ligase (Promega Corporation, Wisconsin, USA). PCR was performed using ligated DNA and primer sequences were designed based on the partial gene sequence of ApEGF, as shown in Table 2. Amplification was performed using forward primer 5′-GAGAGCCCTATCTACAACTGCG-3′ and reverse primer 5′-CACCAAACTACCAACGATTGCA-3′. The PCR amplification conditions consisted of an initial denaturation at 95 °C for 2 min, 30 cycles of denaturation at 95 °C for 60 s, annealing at 55 °C, and extension at 72 °C for 90 s, followed by a final extension step at 72 °C for 5 min. PCR products were cloned into the pGEM-T easy vector to obtain the whole sequence of ApEGF.

### 2.3. Sequencing and Bioinformatics Analyses

For the nucleotide sequencing of the variable regions, pGEM-T-Easy vector clones were purified using a plasmid DNA purification kit (Promega Corporation, Madison, WI, USA). Sequencing was performed using an automatic sequencer (ABI377; PerkinElmer, Waltham, MA, USA) according to the manufacturer’s instructions. A partial gene sequence for ApEGF was obtained. Sequence data were compared using the BLAST program of the National Center for Biotechnology Information (National Institutes of Health, Bethesda, MD, USA). The sequences were aligned using the MEGA 5.1 software. The ExPAsy online tool Compute pI/Mw (https://web.expasy.org/compute_pi/, accessed on 20 July 2018) was used to predict the isoelectric point and molecular weight of the protein. In addition, SignalP (http://www.cbs.dtu.dk/ services/SignalP/, accessed on 20 July 2018) was used to predict protein signal peptides, and the TMHMM server online tool (http://www.cbs.dtu.dk/services/TMHMM/, accessed on 15 May 2019) was used to analyze transmembrane regions.

### 2.4. Quantitative Real-Time PCR

Total cellular RNA was isolated from HaCaT cells using TRI reagent (RNAiso Plus, Takara Bio, Shiga, Japan), according to the manufacturer’s instructions, and reverse transcription was performed by using a cDNA synthesis kit (Takara Bio, Shiga, Japan). After cDNA synthesis, cDNA was amplified by using primer pairs shown in Table 3. Real-time PCR was performed with the FTC3000 real-time PCR system (Funglyn Biotech, TO, Canada) using the SYBR Green PCR kit (Takara Bio, Shiga, Japan), according to the manufacturer’s instructions. Real-time PCR was performed using the following cycling conditions: 4 min at 95 °C, followed by 35 cycles of 94 °C for 20 s, 60 °C for 25 s, and 72 °C for 30 s. Relative mRNA expression levels were expressed as the ratio of the comparative threshold cycle to the expression levels of human β-Actin.

### 2.5. Expression and Purification of Cut-Off ApEGF

First, the cut-off ApEGF gene, in which the two transmembrane domains were cut off, was synthesized. Briefly, the sequence of the cutting-off ApEGF cDNA plasmid containing NcoI and XhoI sites of the pET22b expression vector encoding a 6xHis tag was used to transform *Escherichia coli* Arctic Express. Cells were grown in 500 mL of LB medium containing 50 μg/mL kanamycin at 37 °C with shaking (220 rpm) until the OD600 reached 0.6–0.8. IPTG was added at a final concentration of 0.5 mM, and cells were allowed to grow for an additional 4 h at 37 °C. The cells were then harvested by centrifugation at 6000× *g* for 15 min at 4 °C. The cell pellet was resuspended in 20 mL of cold cell lysis buffer (20 mM Tris-HCl containing 1 mM PMSF and bacterial protease inhibitor cocktail, pH 8.0). The cells were lysed by sonication (power 400 W, 4 s, 8 s intervals, total 20 min). The cell lysate was cleared by centrifugation at 10,000 rpm for 20 min at 4 °C. Inclusion bodies were washed three times with wash solution (20 mM Tris, 1 mM EDTA, 2 M urea, 1 M NaCl, 1% TritonX-100, pH 8.0). According to the proportion, the inclusion bodies were dissolved by lysis buffer (20 mM Tris, 5 mM DTT, 8 M urea, pH 8.0), stored overnight at 4 °C, and then centrifuged at room temperature at 10,000 rpm for 15 min. The protein solution was loaded into a dialysis bag and dialyzed overnight using a buffer (20 mM Tris, 150 mM NaCl buffer, pH 8.0). The extract was slowly loaded onto a Ni-IDA-Sepharose CI-6B (Novagen) bead column, which was equilibrated at 4 °C with column-binding buffer (20 mM Tris, 150 mM NaCl, 5 mM imidazole buffer, pH 8.0 containing 1 mM PMSF and 1 mM DTT). After washing with column buffer (20 mM Tris, 150 mM NaCl, 20 mM imidazole buffer, pH 8.0) until the OD280 was below 0.005, ApEGF was eluted from the Ni-NTA agarose bead column by incubation at room temperature for 5 min with elution buffer (20 mM Tris, 150 mM NaCl, 2500 mM imidazole buffer, pH 8.0 containing 1 mM PMSF and 1 mM DTT). The collected protein solution was added to a dialysis bag and dialysis was performed overnight using a buffer solution (20 mM Tris, 150 mM NaCl buffer, pH 8.0). The concentration of the purified enzyme was determined, and its purity was assessed using SDS-PAGE.

### 2.6. Western Blot

The protein was separated on 12% SDS-polyacrylamide gel and then transferred to PVDF membranes (Biosharp, Beijing, China). Blots were blocked in a 5% skimmed milk powder solution in TBST (1 × TBS and 0.1% Tween 20) for 1 h at 37 °C, washed with TBST, and incubated with the primary antibody (Anti-His- tag:1:1000) overnight at 4 °C. After washing, the membranes were reacted with diluted secondary antibodies (1:5000) for 1 h at room temperature. Immunostained bands were visualized by the enhanced chemiluminescence method.

### 2.7. Cell Culture and In Vitro Scratch Assay

HaCaT cells (a human keratinocyte cell line, the HaCaT cell line was provided by JRDUN Biotechnology Co., Ltd. Shanghai, China) were cultured in Dulbecco’s modified Eagle’s medium containing 10% fetal bovine serum and 1% double antibody (penicillin mixed solution) in a 5% CO_2_ atmosphere at 37 °C. For the in vitro scratch assay, HaCaT cells were seeded in 6-well plates at a density of 8 × 10^5^ cells/well and grown overnight. A marker was used to draw a line at the bottom of the plate. A scratch was then made using a sterile 10-μL pipette tip, and the cells were re-washed. Transforming growth factor (TGF)-β1 (1 ng/mL) was used as a positive control, and a candidate drug was then added to the medium. The scratches were photographed at ×100 under a microscope (XDS-500C, Shanghai, China) at the time indicated after scratching. Wound distance was measured at three points—long, intermediate, and short distances—and presented as mean values.

### 2.8. Analysis of In Vivo Wound Healing

Twenty male SD rats (male, weighing approximately 180–200 g) were purchased from Shanghai Lab. Animal Research Center (Shanghai, China). After 1 week of adaptive feeding, the rats were anesthetized with ether, the back hair of the rats was shaved with a shaver, and the skin was wiped with alcohol. A 1 × 1 cm circular wound was cut behind the skin on the back of the rat. To prevent the contraction of the skin from affecting the experimental results, a rubber ring was sewn around the wound to fix the surrounding skin. The animals were divided into two groups (n = 10 each): the normal group (0.9% NaCl 0.2 mL) and the experimental group (30 µg/0.2 mL cut-off EGF of *A. pernyi*). The drug was administered every morning and afternoon, and wound photographs were taken every 3 days. During the fourth and seventh days of the operation, three rats from each group were sacrificed after cervical dislocation. The unhealed skin of the wound was fixed, washed, dehydrated, transparent, waxed, embedded, and sliced for hematoxylin-eosin staining.

### 2.9. Statistical Analysis

All experiments were performed in triplicate. Data are reported as the mean ± standard deviation. All statistical analyses were performed using the SPSS software (version 17.0; SPSS Inc., Chicago, IL, USA). The statistical significance of differences between the mean values was determined by a one-way analysis of variance followed by Student’s *t*-test. Statistical significance was set at *p* < 0.05.

## 3. Results and Discussion

Thus far, most wound-healing studies have used EGF originating from mammals. There have been few reports of EGF from invertebrates, and no related studies on ApEGF have been conducted. In this study, we performed the sequence, cloning, transcriptional change, and cDNA prokaryotic expression analyses of an EGF gene and examined the wound-healing effect of ApEGF in vitro and in vivo. First, the genomic DNA of the fifth instar larvae of *A. pernyi* was isolated and extracted (Appendix A). Next, the gene sequence of EGF was obtained from the NCBI database, and primers were designed for PCR amplification, as shown in Table 1. However, the primers designed at both ends of the EGF gene did not amplify the fragments. PCR fragments appeared when amplified with primers designed for the 35–432 (forward: 5′ GCGGTCATGTCATCGCCTTTCG-3′; reverse: 5′-GTCCTTGAACTCGCACCGC-3′) base portion of the BmEGF gene. Primers were designed for the obtained EGF sequence, as shown in Table 2, and inverse PCR was performed to obtain the full-length gene sequence of ApEGF (Figure 1). The gene sequence fragment of EGF is 666 bp in length, encoding 221 amino acids, with a predicted protein mass of 24.19 kD, an isoelectric point of 5.15, no signal peptide sequence, and two transmembrane regions.

The protein homology analysis of ApEGF was performed. The ApEGF homologous protein was searched using BLAST. Sequence homology analysis showed 86.1% sequence homology with Bombyx mori (XP_004926209.1), 92.7% with *Manducal sexta* (XP_030025607.1), 92.6% with *Trichoplusia ni* (XP_026739166.1), and 91.8% with *Helicoverpa armigera* (XP_021192633.1), as shown in Figure 2.

The cloning and purification of truncated ApEGF were performed, and the results are shown in Figure 3 and Figure 4 (or Appendix A). ApEGF was cloned into the pET-28a (+), pET-22b, PGEX-2T-1, and other vectors. Proteins were induced with IPTG after the transformation of *E. coli* BL21 (DE3) cells. However, these recombinant ApEGF plasmid DNAs were not expressed in *E. coli*. Thus, we obtained the target protein for intracellular expression using a baculovirus–insect cell protein expression system. The BACI-A expression plasmid was constructed by gene synthesis, and recombinant Bacmid was obtained by blue-white screening after transformation. After identification by PCR, SF9 cells were transfected to obtain the P1 and P2 viruses. Small-scale expression and Western blotting were used to detect the protein expression. Through Western blotting analysis, the protein expression was detected in SF9 cells. The target position was then banded. After lysis and purification, the target proteins were neither enriched nor purified. ApEGF has two transmembrane domains, making it difficult to isolate and purify the protein. Prokaryotic expression and baculovirus insect cell expression were tested, but the protein was not successfully isolated or purified. Therefore, we cut off the two transmembrane regions and then isolated and purified them after prokaryotic expression. The amino acid regions shown in gray in Figure 1 are the two transmembrane regions, and the amino acid sequences of the two transmembrane regions are SLVVWWAVVSAAWALAGACSSSIS and TASIAGGATVAVFLAILVCFGAWV. After truncation, the molecular weight (Mw) of ApEGF was 19.65 KDa and the isoelectric point (PI) was 9.99. The molecular formula was C855H1379N267O244S11. The truncated ApEGF gene was cloned into pET-22b, and the plasmid DNA was transformed into *E. coli* Arctic Express. The IPTG-induced truncation of the ApEGF vector fusion protein was conducted. As shown in Figure 3, no bands of this protein appeared before and after induction, and no bands appeared in the supernatant of disrupted cells after induction, but bands appeared in the precipitate after induction. This indicates that the target protein was mainly present in the precipitate.

The inclusion bodies were renatured, and the target protein was re-dissolved. Next, the target protein was obtained by Ni column affinity purification and then subjected to SDS-PAGE and Western blotting assay. As shown in Figure 4, a fine target band of approximately 20 KDa appeared after the induced bacteria were post-processed. No band was observed in the effluent after Ni was applied to the column, indicating that the protein was successfully purified. Protein purification was further verified by Western blotting.

The effects of truncated ApEGF on the mRNA expression of LMO2, GATA2, RUNX1, SCL, KDR, and C-MYB, which can influence intracellular concentrations in HaCaT cells, were investigated (Appendix A). The real-time PCR data revealed that truncated ApEGF significantly suppressed the mRNA levels of GATA2. Truncated ApEGF significantly increased LMO2 and KDR expression. Especially, KDR expression was about 4.8-times increased compared to the control. RUNX1, SCL, and C-MYB expression increased only slightly. These results showed that truncated ApEGF could be used in wound-healing therapy without scar formation.

Re-epithelialization is a necessary element in the wound-healing process. Wound healing is a complicated process consisting of clot formation, granulation tissue accumulation, inflammatory responses, remodeling, and extracellular matrix deposition [28]. In addition to several mediators released from fibroblasts, inflammatory cells, and keratinocytes, cellular interactions in the dermis and epidermis play a crucial role in the wound-healing process [29,30]. Various factors such as cytokines, growth factors, metalloproteinases, and extracellular matrix proteins affect the wound-healing process, and some exogenous applications of these factors have been verified to support the wound-regeneration process [31,32]. To study the effect of truncated ApEGF on wound healing in vitro, scratches were made in HaCaT cell cultures, which were then allowed to re-epithelialize for 48 h at 37 °C in the presence or absence of ApEGF. Photographs were taken at 0 and 48 h after the scratch was made. We freeze-dried the purified ApEGF protein and used it as the experimental group, with TGF-β1 as the positive control. Cell growth after drug treatment is presented in Figure 5A. Cell growth in the TGF-β1 and truncated ApEGF groups were significantly improved compared with that in the control group. The wound-healing efficiency in the control, TGF-β1, and truncated ApEGF groups is shown in Figure 5B. The truncated ApEGF group showed 211.15% cell growth. The TGF-β1 group, as a positive control group, exhibited 187.32% cell growth.

To evaluate the in vivo wound-healing effect of truncated ApEGF, especially its effect on re-epithelialization, we performed a perforated biopsy to cause skin defects on the backs of rats, and pictures were taken every 3 days (Figure 6). On days 4 and 7, the wound tissues of the rats were fixed, embedded in wax, and stained with hematoxylin and eosin. Photographs of the wound-healing process in the control and truncated ApEGF groups on days 0, 3, 6, 9, 12, and 15 are presented in Figure 7. Compared to the control group, the truncated ApEGF group exhibited significant wound healing on day 3, and by day 15, the wounds in the truncated ApEGF group had fully healed. Therefore, the wound-healing effect of truncated ApEGF was significantly higher than that of the control. The results of hematoxylin and eosin staining analysis of the wound-healing tissue are shown in Figure 7. On days 4 and 7, the rats were sacrificed by neck removal, and the wound-healing tissues were fixed, embedded, and stained with hematoxylin and eosin. On day 4, the wound areas in the ApEGF group were significantly smaller than those in the control group. On day 7, the wounds in the ApEGF group showed many pores compared with those in the control group. This finding shows that truncated ApEGF greatly improves the wound-healing process in rats.

## 4. Conclusions

In this study, we investigated the wound-healing effect of the EGF protein in cells in vitro and rat epidermis in vivo. The growth of HaCaT cells exposed to truncated ApEGF was significantly improved, leading to an accelerated wound closure rate after 48 h compared with the control cells. In particular, the cells treated with truncated ApEGF were about 12.3% increased compared to TGF-β1 as the positive control. These findings suggest that the truncated ApEGF protein greatly improved wound healing. The extent of the re-epithelialization of wounds in rat epidermis treated with truncated ApEGF was also significantly higher than that in the control group. These results indicate that truncated ApEGF can be used in the treatment of wounds, ulcers, burns, scalds, skin transplantation, and cosmetic surgery, and has a wide range of uses in cosmetics and medicine. However, further clinical trials are required.

## Figures and Tables

**Figure 1 insects-13-00975-f001:**
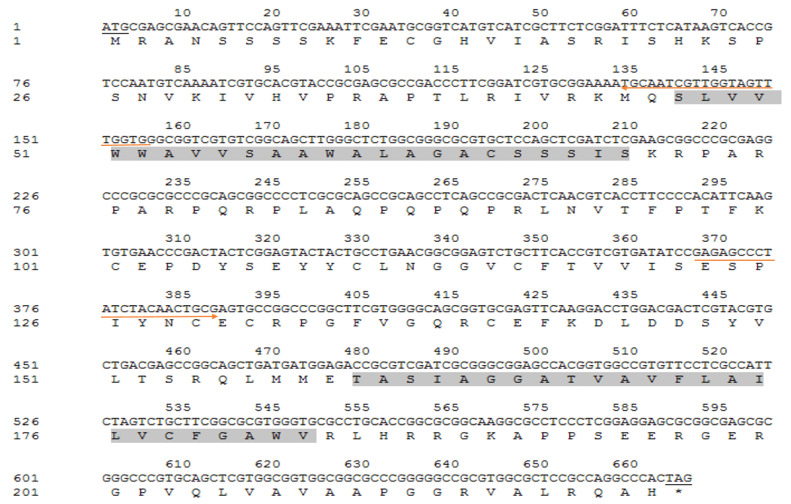
Nucleotide sequence and deduced amino acids of ApEGF. The underlined sequence indicates the start codon and stop codon, and light gray indicates the two transmembrane domains. The underlined sequence indicates the start codon and stop codon, and light gray indicates the two transmembrane domains. The red arrow pointing to the right (→) indicates the forward primer sequence and pointing to the left (←) indicates the reverse primer sequence.

**Figure 2 insects-13-00975-f002:**
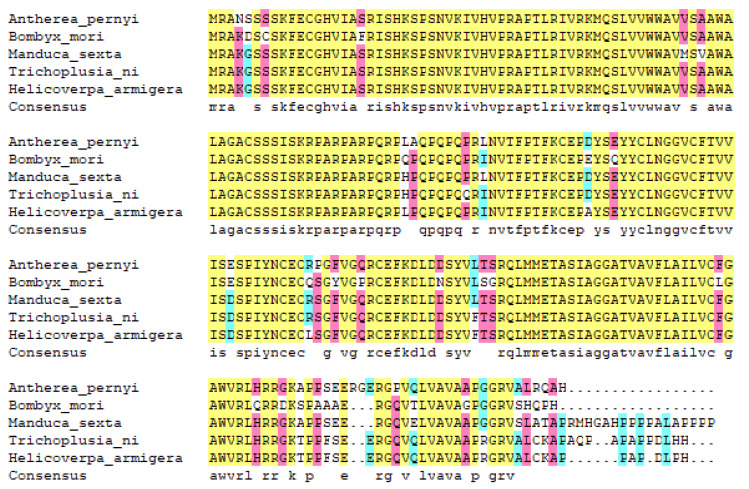
Alignment of amino acid sequences coded by ApEGF and other species. The sequences are as follows: *Antheraea pernyi* (OP407849), *Bombyx mori* (XP_004926209.1), *Manduca sexta* (XP_030025607.1), *Trichoplusia ni* (XP_026739166.1), and *Helicoverpa armigera* (XP_021192633.1).

**Figure 3 insects-13-00975-f003:**
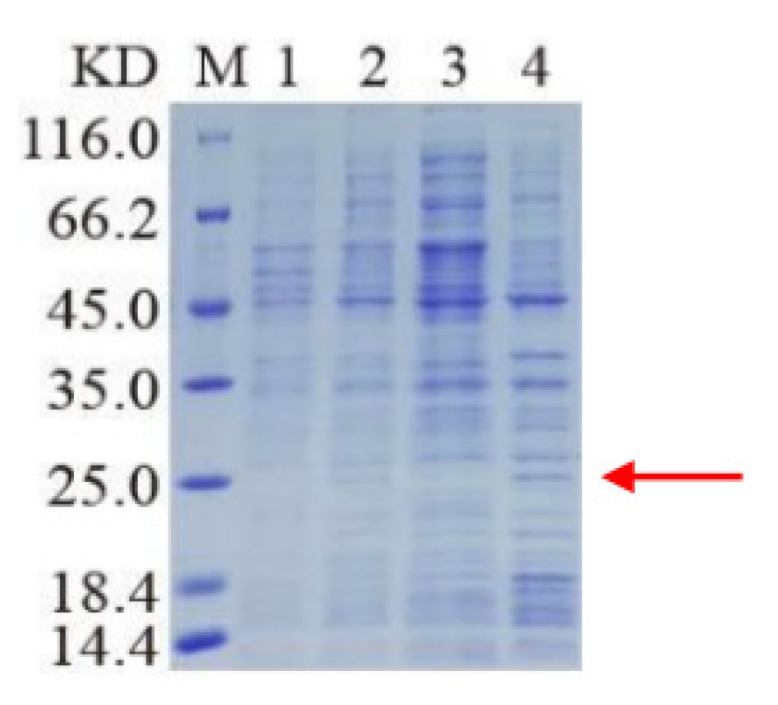
SDS-PAGE analysis of cut-off ApEGF expression. The cut-off ApEGF gene was cloned into pET-22b expression vector encoding a 6 x His tag, and the plasmid DNA was transformed into *E. coli* Arctic Express. The truncated protein ApEGF molecular weight is 19.65 KDa. Lane M: marker; lane 1: bacterial liquid without added inducer; lane 2: bacterial liquid with added inducer; lane 3: broken supernatant after induction; lane 4: broken precipitation after induction. Red arrow indicates target protein.

**Figure 4 insects-13-00975-f004:**
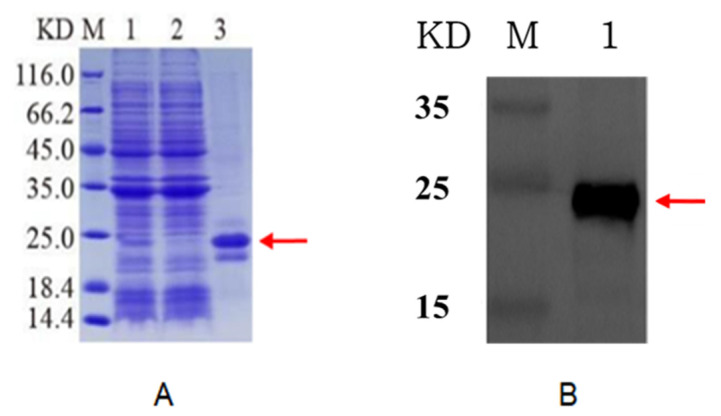
SDS-PAGE analysis of cut-off purified ApEGF with pET-22b expression vector encoding a 6 x His tag. (**A**): M, marker; 1, broken precipitation; 2, effluent; 3, purified protein. (**B**): Western blot analysis of purified cut-off ApEGF with anti-His tag. Lane M: marker; lane 1: purified protein. Red arrow indicates target protein.

**Figure 5 insects-13-00975-f005:**
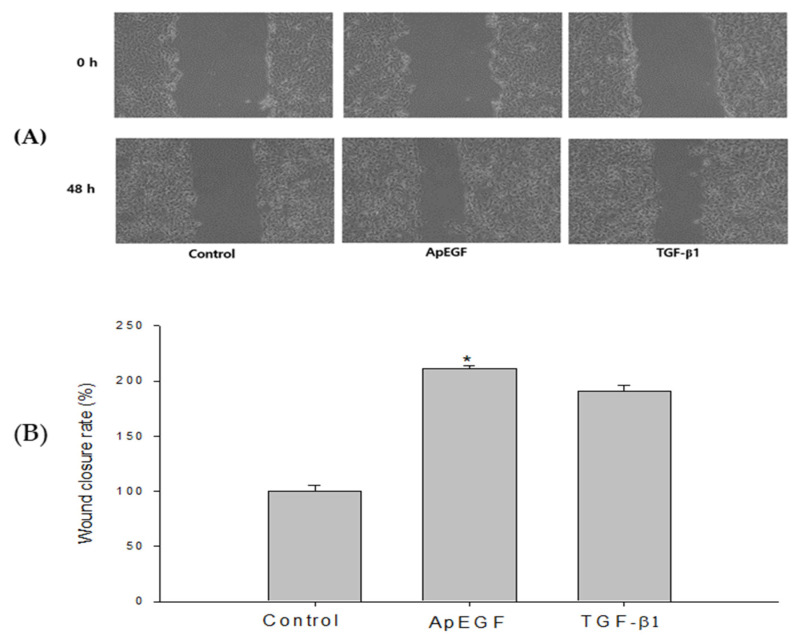
Wound-healing efficiency of the control, truncated ApEGF, and TGF-β1 in vitro. (**A**) Gross wound-size changes during wound healing in each group. Scratch-wound healing of HaCaT confluent monolayers. HaCaT cells were seeded in 6-well plates at a density of 8 × 10^5^ cells/well and maintained in DMEM medium supplemented with 10% fetal bovine serum containing 5% CO_2_ at 37 °C. The picture was taken at 0 h and 48 h after treatment with truncated ApEGF or TGF-β1. (**B**) Longitudinal analysis of wound healing in each group. When the round scratch was performed, the percentage by control was calculated by measuring the narrowed average distance of 48 h. *, *p* < 0.05, statistically significant versus the control.

**Figure 6 insects-13-00975-f006:**
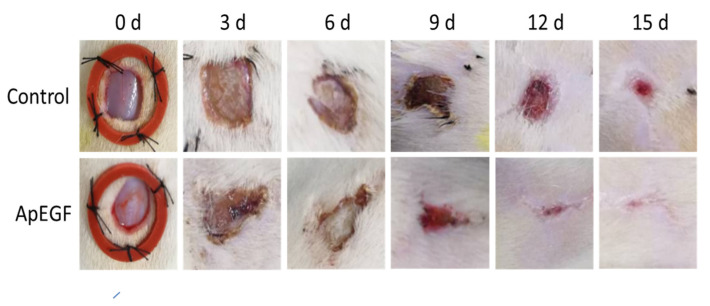
Wound-healing effect of the control and truncated ApEGF in vivo. A 1 × 1 cm circular wound was cut behind the skin on the back of the SD rat. The animals were divided into two groups, the normal group (0.9% NaCl 0.2 mL) and the experimental group (30 µg/0.2 mL cut-off EGF of *A. pernyi*). The drug was administered every morning and afternoon, and wound photographs were taken every 3 days.

**Figure 7 insects-13-00975-f007:**
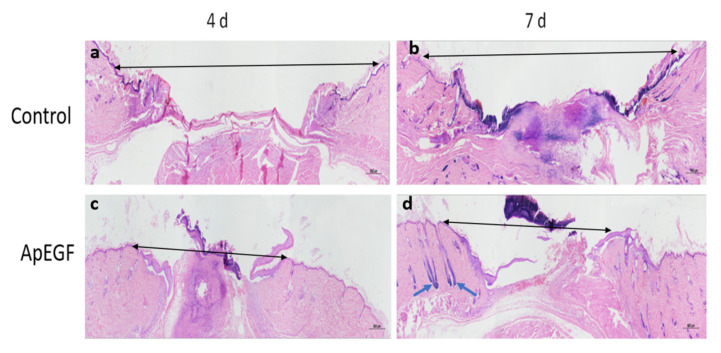
Hematoxylin and eosin staining analysis of round-wound healing in mouse models at 4 and 7 days after treatment with the control and truncated ApEGF. (**a**): HE-staining image of the blank group on the fourth day of wound treatment; (**b**): HE-staining image of the blank group on the seventh day of wound treatment; (**c**): HE-staining image on the fourth day of wound treatment with truncated ApEGF; (**d**): HE-staining image on the seventh day of wound treatment with truncated ApEGF. Black arrows indicate wound distance; green arrows indicate hair follicles.

**Table 1 insects-13-00975-t001:** List of primers for PCR amplification.

Primer Mame	Sequences (5′-3′)
BmEGF-S	TGCGAGCAAAGGATTCCTG
BmEGF-AS	GTGCGGCTGGTGCGAAACGCG
BmEGF-35-F	GCGGTCATGTCATCGCCTTTCG
BmEGF-432-R	GTCCTTGAACTCGCACCGC
BmEGF-69-F	GTCACCGTCCAATGTCAAAATCGTG
BmEGF-560-R	CGCTGCAGCCGCACCCACGC
BmEGF-199-F	AGCTCCATCTCCAAGCGGCC
BmEGF-557-R	TGCAGCCGCACCCACGCTCCGA

**Table 2 insects-13-00975-t002:** List of primers for inverse PCR.

Primer Mame	Sequences (5′-3′)
ApEGF-S1	GAGAGCCCTATCTACAACTGCG
ApEGF-AS1	CACCAAACTACCAACGATTGCA
ApEGF-S2	GCAGCGGTGCGAGTTCAAGGAC
ApEGF-AS2	CTGGAGCACGCGCCCGCCAGAG
ApEGF-S3	AGTGCCGGCCCGGCTCCGTG
ApEGF-AS3	CGCCAGAGCCCAAGCTGCCGAC

**Table 3 insects-13-00975-t003:** List of primers for real-time PCR.

Genes Name		Sequences (5′-3′)
β-Actin	Forward	GACTATGACTTAGTTGCGTTA
	Reverse	GTTGAACTCTCTACATACTTCCG
LMO2	Forward	ATGACAATGCGGGTGAAAGAC
	Reverse	CATCCCATTGATCTTAGTCCACTC
GATA2	Forward	GATGAATGGGCAGAACCGAC
	Reverse	AGACAGGGTCCCCGTTGG
RUNX1	Forward	CAAACCCACCGCAAGTCG
	Reverse	CCGCTCGGAAAAGGACAAG
SCL	Forward	AAGTTGTGCGGCGTATCTTC
	Reverse	ATTCTTGCTGAGCTTCTTGTCC
KDR	Forward	GTGAGCAAAGGGTGGAGGTG
	Reverse	AACATAGACATAAATGACCGAGGC
MYB	Forward	TGATGGGTTTTGCTCAGGC
	Reverse	CATGTAACGCTACAGGGTATGGA

## Data Availability

All data are available in this paper.

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
