# Peer review of "Wound-Healing Effect of Antheraea pernyi Epidermal Growth Factor"

_insects, 2022, doi:10.3390/insects13110975_

Round 1
Reviewer 1 Report
In this study, a gene from Antheraea pernyi named growth factor (ApEGF) was studied. Purified proteins have positive effects in wound healing both in vitro and in vivo. This is a good work that finds the function of EGF in A. pernyi. Some paragraph in this manuscript is not very standard in written also need some modifications. There are some questions need clarify.
1. The results can support the main topic in this study, but the article written style were not common as others. The primers design, gene bioinformatic analysis need brief writing. In the introduction, some plants can heal wound have been introduced (what’s the connection to the topic need reflect), but what is EGF, have not introduced. This will made readers confusion. All about EGF were introduced in discussion.
2. Two important articles about A. pernyi silk fibron wound healing need introduced. (Biomineralization directed by prenucleated calcium and phosphorus nanoclusters improving mechanical properties and osteogenic potential of Antheraea pernyi silk fibroin-based artificial periosteum, Advanced Healthcare Materials, 2021, 2001695. Polydopamine-coated Antheraea pernyi (A. pernyi) silk fibroin films promote cell adhesion and wound healing in skin tissue repair”. ACS Applied Materials & Interfaces, 2019, 11, 38, 34736-34743)
3. Is the genome data of A. pernyi have been used? There may be have the sequence of EGF. Why extract DNA, the prokaryotic expression is using RNA.
4. Some gene analysis figures can be added in supplement data. About the gene qPCR, the GATA, LMO2, KDR should introduced, why detect them. GATA2 (do not have *, P>0.05), can not say significantly suppressed the mRNA level.
5. Conclusion should rewrite, is not the repetition of results.
Author Response
Thank you for reviewing the manuscript. As the reviewer pointed out, we revised them and marked it in red. In this study, a gene from Antheraea pernyi named growth factor (ApEGF) was studied. Purified proteins have positive effects in wound healing both in vitro and in vivo. This is a good work that finds the function of EGF in A. pernyi. Some paragraph in this manuscript is not very standard in written also need some modifications. There are some questions need clarify. 1. The results can support the main topic in this study, but the article written style were not common as others. The primers design, gene bioinformatic analysis need brief writing. In the introduction, some plants can heal wound have been introduced (what’s the connection to the topic need reflect), but what is EGF, have not introduced. This will made readers confusion. All about EGF were introduced in discussion. -->This study is an experiment on the word healing effect using ApEGF as a biomaterial. So I introduced various biomaterials to the introduction. As you pointed out, we moved the EGFs introduced in the discussion to the introduction as follows. EGF is a growth factor derived from single cells, macrophages, and platelets [13], and it plays an important role in various physiological processes [14]. It promotes the proliferation of skin, stomach, lung, tracheal epithelial, and corneal epithelial tissues; accelerates the repair of injured epidermal cells and gastric ulcers; and inhibits gastric acid secretion [15]. EGF can significantly promote the proliferation of fibroblasts as well as the migration and differentiation of epithelial cells in the wound healing process, thereby promoting the re-epithelialization of granulation tissue and wounds [16]. Many researchers have investigated whether EGF can shorten wound healing time. Epstein et al. suggested that EGF increased tensile strength and decreased adverse tissue events, such as failure to produce extracellular matrix proteins in saliva and the severity of oral mucositis during oropharyngeal radiation therapy [17]. Combined treatment with erythropoietin and recombinant human EGF on full-thickness wound healing in a diabetic rat model significantly improved healing time by 50% compared with EGF alone. This result indicated that the combination treatment improved wound healing, possibly through the synergistic action of both growth factors [18]. To evaluate bifunctional fusion proteins as epidermal cell-healing agents, abundant proteins in epidermal cells and fused human EGF to the collagen-binding domain of Escherichia coli were examined for collagen-binding activities, cell proliferation, and phosphorylation signals [19]. Kang et al. [20] developed an EGF via fusion of the skin macromolecule transduction domain (MTD) to improve the percutaneous transmission efficiency of EGF, and the resultant EGF showed excellent permeability to the dermis. These results show that MTD-EGF efficiently improves the existing physical percutaneous transmission method through percutaneous transmission as a raw material for cosmetics containing various active substances. 2. Two important articles about A. pernyi silk fibron wound healing need introduced. (Biomineralization directed by prenucleated calcium and phosphorus nanoclusters improving mechanical properties and osteogenic potential of Antheraea pernyi silk fibroin-based artificial periosteum, Advanced Healthcare Materials, 2021, 2001695. Polydopamine-coated Antheraea pernyi (A. pernyi) silk fibroin films promote cell adhesion and wound healing in skin tissue repair”. ACS Applied Materials & Interfaces, 2019, 11, 38, 34736-34743 --> As point out by reviewer, we inserted two articles about Antheraea pernyi Silk Fibroin on wound healing. Shuai et al designed the biomineralization process induced by prenuclear calcium and phosphorus nanoclusters to nucleate and generate hydroxyapatite crystals on the surface of the A. pernyi fibroin membrane. This result significantly promoted osteogenic differentiation of MSCs in the absence of osteogenic inducer and demonstrate that bone-related matrix proteins are highly expressed (25). Polydopamine-coated A. pernyi silk fibroin films promoted cell adhesion and wound healing in skin tissue repair (26). 25. Shuai, Y.; Lu, H.; Lv, R.; Wang, J.; Wan, Q.; Mao, C.; Yang, M. Biomineralization directed by prenucleated calcium and phosphorus nanoclusters improving mechanicalp Properties and osteogenic potential of Antheraea pernyi silk fibroin-based artificial periosteum, Advan. Healthcare Mater, 2021, 10, 1-13. 26. Wang, J.; Chen, Y.; Zhou, G.; Chen, Y.; Mao, C.; Yang, M. Polydopamine-coated Antheraea pernyi (A.pernyi) silk fibroin films promote cell adhesion and wound healing in skin tissue repair, ACS Appl. Mater. Interfaces, 2019, 11, 34736–34743 3. Is the genome data of A. pernyi have been used? There may be have the sequence of EGF. Why extract DNA, the prokaryotic expression is using RNA. --> Genome data is not used. As far as I know, the experiment was conducted between 2017 and 2019, and the Fernyi genome was not disclosed. In this study, an experiment was conducted using both DNA and RNA samples. However, when RNA samples were used, only partial sequence information was obtained, and the overall sequence was not obtained as RNA samples. When genome DNA was extracted and used, the overall sequence information was obtained. 4. Some gene analysis figures can be added in supplement data. About the gene qPCR, the GATA, LMO2, KDR should introduced, why detect them. GATA2 (do not have *, P>0.05), can not say significantly suppressed the mRNA level. --> We added it in supplement data, as the reviewer pointed out .5. Conclusion should rewrite, is not the repetition of results. -->we rewrote the result part as the reviewer pointed out.
Reviewer 2 Report
The manuscript entitled “Wound Healing Effect of Antheraea pernyi Epidermal Growth 2 Factor” is an interesting study on wound healing using insect agent, the authors cloned the ApEGF gene using inverse PCR from gDNA and expressed cut-off ApEGF using pET22b in E. coli, and purified the recombinant EGF from inclusion bodies, and in vitro and in vivo assays showed that the purified EGF could promote the wound healing. This study may provide a new candidate for wound healing materials from insects and enrich the wound healing agents. I have some comments that might improve the manuscript quality before it could be accepted, listed below:
1. The introduction is not well arranged, and the insect EGF background is almost missing, and there are so many abbreviations without full name and explanation, that will make readers hard to understand. Line 209-233 should be in the introduction part.
2. Line 94, please introduce some background of EGF before you tell the readers what you will do in the study and explain why you choose EGF for further investigation.
3. All the sequence information such as gene access number is missing, I didn’t find the ApEGF information in NCBI database, did you deposit the ApEGF in database? If not, please submit the sequence to database and put the acc number in your manuscript. And the acc number in figure 2 is also missing.
4. Could you please include the clone results such as the gel images with the target bands in the supplementary? It will be better if you add a diagram that shows how did you do the inverse PCR and clone the full length of ApEGF.
5. In figure 2, please don’t hide the identical sequence, please use * to show the identical sequence. And I believe the EGF family is conserved, you use the recombinant EGF from insect in rat wound healing experiments, so you can also include a rat EGF sequence in figure 2, if the EGF family is conserved from insect to rat, that will make sense why you used insect EGF in rat wound healing experiments.
6. Please add more details in the plasmid and bacmid construction, such as the restriction enzyme sites and primers used. When you construct the bacmid, did you add a signal peptide in the ApEGF sequence? Based on your signal peptide analysis, there is no SP in ApEGF, if the EGF was tagged with a SP, the EGF should be secreted into medium, which might be helpful for purification. Anyway, you have already expressed and purified the cut-off ApEGF using pET-22b in E. coli. If you would like to express full length of EGF in Sf9, the wound healing assay will be perfect. You know the cut off ApEGF looks like a new protein.
7. You used a lot of vectors for the EGF expression, but in line 260-282, the details are not enough and easy to make readers confused, please list them in a table, and add more details such as vector names the restriction enzyme sits, primers used for plasmid construction in the table, and which vector you used for full length EGF expression, which one you used for cut-off EGF expression, that will make the method clear.
8. Line 278, can’t understand “The molecular formula was C855……..S11”.
9. Please add more details in all the Figure legends, don’t use title for everything, limited information makes readers confused and takes longer time to figure out what’s the points in the figures. For example, in figure 3 and 4, you should read the content on page7 to figure out which plasmid you used, and which antibody you used for WB? WB method is missing. In figure 6, please tell readers which cell line you used for the assay and how you analyze the results in the figure legend.
10. Line 299, please cite ref. to explain why you check the expression of these genes expression in HaCaT cells.
11. In Figure 7 and figure 8, the results were not well deciphered, please rewrite this part and well explain your results, tell readers why you use this method, how to understand your results, based on your figure legends in figure 7 and 8, it’s hard to understand what you are saying. And for the figure 7, why the scar got red on day 9-15? Are these wounds from the same rat, what’s the scale in figure 7?
12. The manuscript was not written based on the Insects template, please download latest articles from Insects, and rearrange the structure.
13. Please check the font size, line 136, 174, 389.
14. Line 240, the number of primers is not right, please correct it.
15. Line 242, please change reverse PCR to inverse PCR.
16. Pay attention to your statistical analysis, choose the right one for your analysis, for figure 5 and 6, t-test is OK.
17. Could you mark the primers you used for the clone in figure 1?
18. The vector name PET22b should be pET-22b, and please check all the vector names.
Author Response
Open Review 2. Thank you for reviewing the manuscript. As the reviewer pointed out, we revised them and marked it in red. Comments and Suggestions for Authors The manuscript entitled “Wound Healing Effect of Antheraea pernyi Epidermal Growth 2 Factor” is an interesting study on wound healing using insect agent, the authors cloned the ApEGF gene using inverse PCR from gDNA and expressed cut-off ApEGF using pET22b in E. coli, and purified the recombinant EGF from inclusion bodies, and in vitro and in vivo assays showed that the purified EGF could promote the wound healing. This study may provide a new candidate for wound healing materials from insects and enrich the wound healing agents. I have some comments that might improve the manuscript quality before it could be accepted, listed below: 1. The introduction is not well arranged, and the insect EGF background is almost missing, and there are so many abbreviations without full name and explanation, that will make readers hard to understand. Line 209-233 should be in the introduction part. -> We rearranged the introduction part. There's no insect EGF background for Wound healing effect. Full name inserted successfully. And we moved Line 209-233 to in the introduction part. 2.Line 94, please introduce some background of EGF before you tell the readers what you will do in the study and explain why you choose EGF for further investigation. As reviewer pointed out, I introduced the background of EGF in the introduction and also introduced the reason why I chose EGF. 3.All the sequence information such as gene access number is missing, I didn’t find the ApEGF information in NCBI database, did you deposit the ApEGF in database? If not, please submit the sequence to database and put the acc number in your manuscript. And the acc number in figure 2 is also missing. -> As reviewer pointed out, we inserted the gene access number as follows The gene sequence fragment of EGF is 666bp in length, encoding 221 amino acids (GenBank Accession No. OP407849), Sequence homology analysis showed 86.1% sequence homology with Bombyx mori(XP_004926209.1), 92.7% with Manducal sexta (XP_030025607.1), 92.6% with Trichoplusia ni(XP_026739166.1), and 91.8% with Helicoverpa armigera (XP_021192633.1) as shown in Fig. 2. 4.Could you please include the clone results such as the gel images with the target bands in the supplementary? It will be better if you add a diagram that shows how did you do the inverse PCR and clone the full length of ApEGF. -> As you pointed out, we included the above in the supplementary 1. 5.In figure 2, please don’t hide the identical sequence, please use * to show the identical sequence. And I believe the EGF family is conserved, you use the recombinant EGF from insect in rat wound healing experiments, so you can also include a rat EGF sequence in figure 2, if the EGF family is conserved from insect to rat, that will make sense why you used insect EGF in rat wound healing experiments. -> As reviewer pointed out, we modified Figure 2. However, when comparing the EGF amino acid sequences in mice, the homogeneity is too low to compare the EGF in mice and insects. Therefore, the mechanism for wound healing by EGF in insects is thought to need to be studied further. 6.Please add more details in the plasmid and bacmid construction, such as the restriction enzyme sites and primers used. When you construct the bacmid, did you add a signal peptide in the ApEGF sequence? Based on your signal peptide analysis, there is no SP in ApEGF, if the EGF was tagged with a SP, the EGF should be secreted into medium, which might be helpful for purification. Anyway, you have already expressed and purified the cut-off ApEGF using pET-22b in E. coli. If you would like to express full length of EGF in Sf9, the wound healing assay will be perfect. You know the cut off ApEGF looks like a new protein. -> As reviewer pointed out, we added more details in the plasmid and bacmid construction, such as the restriction enzyme sites and primers used. This protein is a protein that does not have a signal peptide and is not a secreted protein. The protein used in this study was membrane protein, and protein purificatian was attempted with SF cell line, but it was not successful. So we cut off the transmembrane regions, and then we purify the protein, and we cut off the transmembrane regions, so we can see it as a new protein. 7.You used a lot of vectors for the EGF expression, but in line 260-282, the details are not enough and easy to make readers confused, please list them in a table, and add more details such as vector names the restriction enzyme sits, primers used for plasmid construction in the table, and which vector you used for full length EGF expression, which one you used for cut-off EGF expression, that will make the method clear. -> We're not sure what the problem is. The entire length of protein is a membrane protein, and it was explained that it was purification by cutting off the membrane part because it was not obtained by purification using various methods. And we don't really understand what to tabulate. we don't think there's any part that can be made into a table. We would appreciate it if you could explain it in detail. If possible, we would like to receive the related reference. 8.Line 278, can’t understand “The molecular formula was C855……..S11”. -> The ExPASy - ProtParam tool was used to obtain basic protein information such as the size, isoelectric point PI, and molecular weight of the cut protein. 9.Please add more details in all the Figure legends, don’t use title for everything, limited information makes readers confused and takes longer time to figure out what’s the points in the figures. For example, in figure 3 and 4, you should read the content on page7 to figure out which plasmid you used, and which antibody you used for Western blot (WB)? WB method is missing. In figure 6, please tell readers which cell line you used for the assay and how you analyze the results in the figure legend. We added more details in all the Figure legends and Western blot method Fig. 1: Nucleotide sequence and deduced amino acid of ApEGF. The underlined sequence indicates the start codon and stop codon, and light gray indicates the two transmembrane domains. The red arrow pointing to the right(→) indicates forward primer sequence and pointing to the left (←) indicates reverse primer sequence. Fig.2. Alignment of amino acid sequences coded by ApEGF and other species. The sequences are as follows: Antherea pernyi ( OP407849), Bombyx mori (XP_004926209.1), Manduca sexta ( XP_030025607.1), Trichoplusia ni (XP_026739166.1), and Helicoverpa armigera (XP_021192633.1). Fig. 3. SDS-PAGE analysis of cut-off ApEGF expression. The cut-off ApEGF gene was cloned into pET-22b expression vector encoding a 6xHis tag, and the plasmid DNA was transformed into E. coli Arctic Express. The truncated protein ApEGF molecular weight is 19.65KDa. Lane M: marker; Lane 1: bacterial liquid without added inducer; Lane 2:bacterial liquid with added inducer; Lane 3: broken supernatant after induction; Lane 4:broken precipitation after induction. Fig.4. (A) SDS-PAGE analysis of cut-off purified ApEGF with pET-22b expression vector encoding a 6xHis tag. (A): M, marker; 1, broken precipitation; 2, effluent; 3, purified protein. (B): western blot analysis of purified cut-off ApEGF with anti-His-tag. Lane M: marker; Lane 1: purified protein Fig.5. Wound healing efficiency of the control, truncated ApEGF, and TGF-β1 in vitro. (A)Gross wound size changes during wound healing in each group. Scratch wound healing of HaCaT confluent monolayers. HaCaT cells were seeded in 6-well plates at a density of 8 × 105 cells/well and maintained in DMEM medium supplemented with 10% fetal bovine serum containing 5% CO2 at 37 °C. The picture was taken at 0 h and 48h after treatment of truncated ApEGF, or TGF-β1. (B) Longitudinal analysis of wound healing in each group. When the round scratch was performed, the percentage by control was calculated by measuring the narrowed average distance of 48 hours. p < 0.05, statistically significant versus the control. Fig 6 Wound healing effect of the control and truncated ApEGF in vivo. 1 × 1 cm circular wound was cut behind the skin on the back of the SD rat. The animals were divided into two groups. the normal group (0.9% NaCl 0.2 mL) and the experimental group (30 µg/0.2 mL cut-off EGF of A. pernyi). The drug was administered every morning and afternoon, and wound photographs were taken every 3 days. Fig 7: Hematoxylin and eosin staining analysis of round wound healing in mouse models at 4 and 7 days after treatment with control and truncated ApEGF. a: HE staining image of the blank group on the fourth day of wound treatment; b: HE staining image of the blank group on the seventh day of wound treatment; c: HE staining image on the fourth day of wound treatment with truncated ApEGF; d: HE staining image on the seventh day of wound treatment with truncated ApEGF. 2.6 Western blot The protein was separated on 12% SDS-polyacrylamide gel and then transferred to PVDF membranes (Biosharp, China). Blots were blocked in a 5% skimmed milk powder solution in TBST (1 × TBS and 0.1% Tween 20) for 1 h at 37℃, washed with TBST, and incubated with the primary antibody(Anti-His- tag:1:1000) overnight at 4℃. After washing, the membrane were reacted with diluted secondary antibodies (1:5000) for 1 h at room temperature. Immunostained bands were visualized by the enhanced chemiluminescence method. 10. Line 299, please cite ref. to explain why you check the expression of these genes expression in HaCaT cells. -> As you pointed out, we insert a reference. To study the effect of truncated ApEGF on wound healing in vitro, scratches were made in HaCaT cell cultures [2], which were then allowed to re-epithelialize for 48 h at 37 °C in the presence or absence of ApEGF. Yoon, E. J.; Choi, D. B.; Yu ,I. S.; Cho, H. Synthesis and structure-activity relationship study of 2,4-dioxothiazolidin-5-ylidene derivatives for 15-hydroxyprostaglandin dehydrogenase inhibitory activity, prostaglandin E2 release, and wound healing effect. Biotechnol. Bioprocess Eng. 2021, 26, 933-955. 11.In Figure 7 and figure 8, the results were not well deciphered, please rewrite this part and well explain your results, tell readers why you use this method, how to understand your results, based on your figure legends in figure 7 and 8, it’s hard to understand what you are saying. And for the figure 7, why the scar got red on day 9-15? Are these wounds from the same rat, what’s the scale in figure 7? Day 9-15 scar redness is the healing period of the wound and seems to be reddened by the creation of new tissues and blood vessels. The same rat's wounds are certain. The experiment was conducted with 5 mice in the mediating group. 12.The manuscript was not written based on the Insects template, please download latest articles from Insects, and rearrange the structure. -> As you pointed out, we wrote the manuscript based on the insect template as follows. Shuai et al designed the biomineralization process induced by prenuclear calcium and phosphorus nanoclusters to nucleate and generate hydroxyapatite crystals on the surface of the A. pernyi fibroin membrane. This result significantly promoted osteogenic differentiation of MSCs in the absence of osteogenic inducer and demonstrate that bone-related matrix proteins are highly expressed [25]. Polydopamine-coated A. pernyi silk fibroin films promoted cell adhesion and wound healing in skin tissue repair [26]. 13.Please check the font size, line 136, 174, 389. -> As you pointed out, we checked them. 14.Line 240, the number of primers is not right, please correct it. -> We changed it as follows: Thus, we designed primers for the 35-58 and 414-432 nucleic acids of the silkworm EGF sequence and performed PCR amplification to obtain a part of the EGF sequence 15.Line 242, please change reverse PCR to inverse PCR. -> We changed reverse PCR to inverse PCR in Line 242. 16. Pay attention to your statistical analysis, choose the right one for your analysis, for figure 5 and 6, t-test is OK. ->We modified as follows: The statistical significance of differences between the mean values was determined by one-way analysis of variance followed by student t-test 17.Could you mark the primers you used for the clone in figure 1? -> As you pointed out, we made Table 1 and 2 for the primers used for the clone in figure 1 and 2 18.The vector name PET22b should be pET-22b, and please check all the vector names. -> As you pointed out, we modified the vector name.
